# Advanced In Vitro Models for Preclinical Drug Safety: Recent Progress and Prospects

**DOI:** 10.3390/cimb47010007

**Published:** 2024-12-26

**Authors:** Dileep G. Nair, Ralf Weiskirchen

**Affiliations:** 1Institute of Molecular Pathobiochemistry, Experimental Gene Therapy and Clinical Chemistry (IFMPEGKC), University Hospital Aachen, D-52074 Aachen, Germany; rweiskirchen@ukaachen.de; 2Keliomics Inc., 4640 S Macadam Ave #270, Portland, OR 97239, USA

**Keywords:** OOC, organ-on-a-chip, DILI, drug-induced liver injury, AI/ML, artificial intelligence and machine learning, HTS, high-throughput screening, MSFLD, metabolic dysfunction-associated fatty liver disease (MAFLD), iPSC, induced pluripotent stem cells, MPS, microphysiological systems

## Abstract

The majority of drugs are typically orally administered. The journey from drug discovery to approval is often long and expensive, involving multiple stages. A major challenge in the drug development process is drug-induced liver injury (DILI), a condition that affects the liver, the organ responsible for metabolizing most drugs. Traditionally, identifying DILI risk has been difficult due to the poor correlation between preclinical animal models and in vitro systems. Differences in physiology between humans and animals or cell lines contribute to the failure of many drug programs during clinical trials. The use of advanced in vitro systems that closely mimic human physiology, such as organ-on-a-chip models like gut–liver-on-a-chip, can be crucial in improving drug efficacy while minimizing toxicity. Additionally, the adaptation of these technologies has the potential to significantly reduce both the time and cost associated with obtaining safe drug approvals, all while adhering to the 3Rs principle (replacement, reduction, refinement). In this review, we discuss the significance, current status, and future prospects of advanced platforms, specifically organ-on-a-chip models, in supporting preclinical drug discovery.

## 1. Introduction

The current drug discovery process is costly and plagued by several bottlenecks, including a prolonged timeline and a high failure rate of approximately 90% from Phase 1 trials to market. The primary reasons for this elevated failure rate are the lack of efficacy and toxicity issues that often emerge at later stages of drug development. Drug-induced liver injury (DILI) significantly contributes to these high attrition rates, sometimes leading to the withdrawal of approved drugs [1]. A major challenge in addressing these adverse effects is the inability to identify them using traditional animal models.

There is an urgent demand for efficient, human-relevant in vitro models that can screen drug candidates utilizing human-based liver systems to detect potential hepatotoxicity. These advanced models offer advantages such as high-throughput capability, cost-effectiveness, and reduced resource requirements compared to conventional in vivo approaches, all while adhering to the 3Rs principle (replacement, reduction, refinement).

Approximately 80% of best-selling drugs are orally administered [2]. Therefore, developing a combined ex vivo model that integrates gut and liver systems would be highly beneficial. By incorporating ex vivo models into a microfluidics platform, characterized by ease of control, high resolution, rapid processing, and low cost, it is possible to create a system that enhances throughput and efficiency. Additionally, individual efficient models (either gut or liver) have significant potential for studying disease progression and serving as advanced disease models for drug discovery.

However, optimizing these ex vivo organ systems involves complex procedures with numerous parameters to consider—including media composition, oxygen and nutrient gradients, growth factors/cytokines essential for proper maintenance, shear stress conditions, extracellular matrix composition, and rigidity. The integration of artificial intelligence (AI) and machine learning (ML) could play a pivotal role in this optimization process. When combined with advanced in vitro models, AI/ML can enhance preclinical studies by addressing factors such as environmental influences, lifestyle variations, genetic diversity, nutrient supply dynamics, and drug metabolism.

## 2. Animal Models for Predicting Toxicity

Animal models have been used to study human physiology and pathophysiology. The progress in drug discovery and development requires advancements in animal models associated with therapeutic areas. The complexity of human physiology can be mimicked to an extent using specialized animal models [3]. Additionally, toxicology studies and studies requiring multiple organ-level manipulations are typically conducted in animal models before advancing to human clinical trials. Some examples of the significance of animal models are described below.

Rat and mice models are the most commonly used animal models to investigate the overall effects of therapeutics or treatments on the body. The availability of extensive databases containing research data from animal studies is also an advantage of animal models. The development of ‘humanized mice’ has emerged as a major tool for accurately modeling human immune responses [4,5,6]. Other commonly used murine model types include genetically engineered mouse models and xenograft models [7]. These models have been successfully utilized to model human immunity, discover and develop therapeutic agents, and conduct toxicology studies. However, these models may have several interspecies incompatibilities of cytokines and cytokine receptors.

Non-human primates, due to their similarity in biochemical and phylogenetic aspects, have been extensively used for vaccine development, orthopedic surgical techniques or devices, and Parkinson’s disease [8]. Studies involving primates can be long-term and have benefitted diseases like Parkinson’s disease. Disease conditions such as human immunodeficiency virus (HIV), Zika virus, and human tuberculosis (TB) are modeled in primates as murine models lack the ability to become infected or mimic human pathology. Most recently, SARS-CoV-2 (COVID-19) research has extensively utilized non-human primates, including rhesus macaques, to understand the disease and develop vaccines and treatment options [9]. The availability of previous research data and the similarity of immune responses to humans make rhesus macaques a reliable model.

Over the past two decades, more than USD 28 billion per year has been spent on irreproducible preclinical research in the United States [10]. Animal testing has traditionally been employed to predict drug safety in humans during preclinical studies. However, this process is often time-consuming and costly, contributing to the high failure rates currently observed in drug development. Additionally, pharmacogenomic differences between humans and animals can lead to discrepancies in drug efficacy and toxicity.

A comprehensive review by Gail Van Norman highlights two critical decisions regarding animal testing of drug candidates: the safe tagging of a toxic drug and the toxic tagging of a beneficial drug [11]. A notable example of a misclassified drug is Vioxx (rofecoxib) from Merck, which was linked to numerous cases of myocardial infarction and stroke, resulting in over USD 8.5 billion in legal settlements [12]. Siramshetty et al. reported that nearly half of the 578 drugs withdrawn or discontinued post-approval in the United States and Europe were due to toxicity issues [13]. Accessing human-relevant data before advancing to clinical trials may provide greater confidence than relying solely on animal testing data. Positive results from preliminary screenings using humanized models can facilitate project progression that might otherwise be terminated based solely on animal data.

In light of these inherent challenges associated with animal-based data, the FDA Modernization Act 2.0 has recently been approved, allowing for alternatives to animal testing for drug and biological product applications [14]. These alternatives include advanced in vitro models such as organoids, OOC systems, and human-induced pluripotent stem cells (iPSCs), as well as AI and ML methods for assessing drug metabolism and toxicity [14]. Similarly, the European Union (EU) has implemented a complete ban on cosmetic products developed using animal models, followed by countries such as Canada, Brazil, Colombia, and Mexico [15]. The growing acceptance of alternative preclinical models has spurred efforts to develop efficient and humanized systems.

Recent advances in in vitro platforms are discussed below with an emphasis on toxicity evaluation—a significant bottleneck in drug discovery.

## 3. In Vitro Cellular Models for Drug Discovery

The process of drug discovery and development is complex, costly, and time-consuming, typically taking between 10 and 15 years and requiring billions of dollars [16] (Figure 1). In vitro cellular models play a crucial role in high-throughput screening (HTS), serving as disease models for efficacy assessments and safety evaluations but requiring further advancements [17]. Over the years, these models have evolved into rapid and reproducible tools for studying both efficacy and toxicity. Below are some of the major applications and advancements in this field. Elfawal et al. recently reported a high-throughput screening platform where over 30,000 compounds were screened to identify 55 compounds with broad anthelmintic activity against gastrointestinal nematodes [18].

### 3.1. In Vitro Cellular Models for High-Throughput Screening

Cell-based in vitro models are widely used for screening and account for nearly half of HTS efforts, especially for orally available drugs. Unlike biochemical or acellular assays, cellular assays reflect molecular interactions within a cellular environment, involving multiple biochemical pathways and their interactions and effects. Furthermore, cellular assays provide deeper insights into toxicity profiles, impacts on various signaling pathways or receptors, and overall cellular effects.

Compared to in vivo counterparts, cell-based in vitro assays offer advantages in scalability, cost-effectiveness, and reproducibility. Establishing effective in vitro assays can also reduce the number of animals used and sacrificed to obtain data related to drug efficacy. By using optimized in vitro screening methods, thousands of compounds can be evaluated using fluorometric, luminescent, and colorimetric platforms across 384-well to 1036-well formats. HTS methodologies typically involve miniaturized assays with minimal reagent requirements and have significant potential for multiparameter measurements from individual wells [19].

However, major challenges include optimizing assay conditions to develop and validate effective protocols that meet statistical acceptance criteria. With advancements in AI, liquid handling technologies, as well as enhanced readouts and data processing capabilities, the development of assays targeting novel pathways or approaches has become increasingly common. Using cell-based HTS platforms allows researchers to screen large numbers of compounds to identify the most active molecules, conduct hit-to-lead studies, and assess potential stability and bioavailability using advanced in vitro models as described in the sections below.

### 3.2. Drug Toxicity Testing Using In Vitro Models

Precision in determining the toxicity of drug candidates is crucial for their development, approval, and ultimate use. Incorporating toxicity assessments during key stages of drug discovery, such as hit-to-lead optimization, structure-activity relationship (SAR) studies, and ADME (absorption, distribution, metabolism, and excretion) processes, can help minimize costs and reduce potential failure risks [20]. Traditional drug toxicology approaches include computational models, biochemical assays, cellular in vitro assays, and in vivo studies. While computational screening is commonly used for initial HTS projects during the drug discovery phase, it may lack comprehensive outcomes on a global physiological scale compared to advanced cellular assays. Specialized cell-based assays are used to identify organ-specific toxicities.

Routine 2D cell cultures can offer basic insights into cytotoxicity. However, they often do not replicate the tissue-specific mechanical and biochemical characteristics of target organs. Current in vitro models for assessing liver hepatotoxicity utilize various cell sources—including genetically engineered cells, hepatoma cell lines, stem cell-derived hepatocyte-like cells, primary hepatocytes, and co-cultures of hepatocytes with non-parenchymal cells (NPCs). While co-cultures of hepatocytes and NPCs provide the most physiologically relevant model, maintaining consistency can be challenging [21]. Integrating a co-culture model into a 3D culture system could enhance predictions of in vivo hepatotoxicity and hepatic clearance of orally available drugs in a high-throughput manner.

To address these challenges, organ-specific cell-based assays, including co-cultures, are increasingly used for HTS assays. However, incorporating more physiological parameters such as extracellular matrix components and organ-specific cell-to-cell interactions can significantly improve the identification of potential toxicity. In both preclinical and clinical phases, toxicology studies focus on risk identification, management, and mitigation. The findings from these studies inform proper clinical trial design and execution.

Identification of an active and selective compound is not enough to consider it a potential candidate for drug development. Poor pharmacokinetics and toxicological profiles are responsible for the majority of drug program failures (63% of all preclinical candidates) [22,23]. Absorption, distribution, metabolism, and excretion (ADME) studies are crucial in drug development, with 40% of drug candidates failing solely due to poor human pharmacokinetics before formal implementation in the drug discovery process [24]. Several ADME assays are available in high-throughput formats and are typically initiated in the early stages of discovery. Some examples include cytochrome inhibition assays using non-fluorescent probe substrates [25] and Caco-2 cell permeability assays using LC-MS (liquid chromatography/mass spectrometry) [26]. Recent approaches to expedite ADME data collection include automated sample preparation, fast chromatography, automated data processing, and sample pooling [23].

Perception of structure–activity relationships (SARs) data is crucial for drug development. Small molecule profiling typically involves characterizing compounds through multiple biological measurements [27]. SAR studies aid in understanding the mechanism of action and provide detailed information on biological effects [27]. Data can be obtained by incorporating information from various assays such as binding studies, cell-line studies for drug sensitivity, and other high-throughput assays [28,29,30]. The availability of public databases like Binding DB, ChEMBL, and ChemBank helps to expedite and streamline the process [31,32]. Furthermore, advancements in high-throughput assay platforms, like high-throughput gene expression signature analysis and multiplex cytological profiling assays for measuring diverse cellular states, have enabled the characterization of SAR analysis in a high-throughput manner [33]. Progress in computational methods also plays a significant role in deducing SAR data from high-dimensional profiling data [27].

Recently, there has been a strong push to establish human-relevant ex vivo models due to the limitations associated with traditional animal models. Some notable advanced in vitro systems include 3D cell culture models such as organoids and OOC or microphysiological systems (MPSs).

### 3.3. Advantages of Complex Cell Culture Models for Drug Discovery and Development

Advanced 3D cell culture models can effectively maintain cell differentiation, as well as RNA and protein expression levels that closely resemble those found in actual organs [34]. These models are widely utilized in drug discovery projects and encompass various formats, including spheroids, organoids, and individual cells suspended within an extracellular matrix [35].

Spheroids are multicellular cultures compatible with a range of cell types, including stem cells, neuronal cells, epithelial cells, and hepatocytes. The advantages of spheroid cultures include their capacity to develop co-cultures and their ability to replicate gradients of nutrients, oxygen, and metabolites. Additionally, they facilitate interactions between cells as well as between cells and the extracellular matrix, mimicking physiological or pathophysiological conditions [36].

Organoids consist of organ-specific cell types and can be classified as tissue-derived or stem cell-derived organoids based on their origin. Stem cells used in organoid formation may be either organ-restricted adult stem cells or pluripotent induced/embryonic stem cells. Generally, organoids contain both stem cells and differentiated cells, allowing them to generate multicellular structures that exhibit tissue-level architecture and function.

Intestinal organoids have significant applications, particularly in oral drug screening and delivery, studying gut microbiome interactions, disease modeling, and developmental research (Figure 2).

One of the earliest studies conducted by Sato et al. in 2009 demonstrated that the leucine-rich repeat-containing G protein-coupled receptor 5 (Lgr5) stem cells possess the ability to develop crypt-villus structures with native tissue-like functions [37]. Qu and his team developed an efficient intestinal organoid model for injury-induced intestinal regeneration, which has the potential to identify candidates that could promote this regeneration [37]. Mature epithelial cell differentiation into enterocytes, goblet cells, Paneth cells, and enteroendocrine cells is associated with the expression of specific markers such as villin, Mucin 2, lysozyme, and chromogranin A, respectively [38].

Liver organoids can be derived from both embryonic or pluripotent stem cells and adult stem cells. Cholangiocyte organoids, which are based on cholangiocytes that line the biliary tree, play a role in parenchymal fibrosis. Huch and his team have successfully established cholangiocyte organoids from both human and mouse sources [39,40,41]. These organoids are valuable for studying the molecular aspects of cholangiopathies and for developing personalized therapeutic strategies.

Hepatocytes constitute the majority of liver cells, and hepatic organoids have been utilized to model various liver diseases [42]. Regenerative hepatocytes resulting from liver injury or other pathophysiological conditions can repopulate to compensate for cell loss. The advantages of liver organoids include their potential for gene editing, biobanking, and HTS. Personalized or patient-centric organoids can be created to recreate specific pathophysiological conditions. Both pluripotent stem cell (PSC)-derived and adult stem cell (ASC)-derived organoids are widely used in various studies [43].

Marsee and her team reported a systematic nomenclature for organoids, including hepatic organoids categorized by their source cells [44]. The secretion of albumin and the activity of cytochromes are commonly used criteria to assess the mature hepatic phenotype. However, one major challenge in establishing these organoids is the lengthy cell differentiation process, along with complex media requirements that can affect the functional lifespan of the models and lead to batch-to-batch variability [45]. Recently, Kamishibahara et al. reported using Laminin-511 (LN) direct coating technology to successfully differentiate human induced pluripotent stem cells (hiPSCs) into hepatocytes, yielding functional hepatic organoids [46]. Nevertheless, more complex systems with more human relevance are needed to detect efficacy and potential toxicity in drug candidates (Figure 3).

### 3.4. Organ-on-a-Chip Platform for Drug Development

Organ-on-a-chip (OOC) technology enables the modeling of complex organs using a microfluidic design that incorporates cellular microenvironments, simulating the physiological, mechanical, and chemical properties of the tissue of origin. This innovative technology has the potential to address many challenges associated with traditional cell culturing and in vivo models while providing greater insights into human physiology at both single-organ and multi-organ levels. Key components involved in OOC systems include microfluidics, cellular microenvironments, cell models, and sensing elements. The successful integration of these components is crucial for the efficiency and robustness of OOC systems.

Since the pioneering work by Huh et al. in 2010 on the ’breathing lung’ model, diverse OOC models for various organs—including lung, liver, brain, gut, and heart—have been reported [47,48]. Multi-organ OOCs that simulate physiological interactions between two or more organs are also being explored. Major applications of OOC technology include drug safety evaluation, pharmacokinetic/pharmacodynamic (PK/PD) modeling, disease modeling, and personalized medicine.

Approximately 80% of drugs are administered orally, with the liver playing a critical role in metabolizing most of these compounds. Traditional 2D cell cultures have been extensively utilized to study drug availability; Caco-2 monolayers are particularly popular due to their simplicity and efficiency in analyzing transepithelial drug transport [49]. The inclusion of a mucus layer is often achieved by co-culturing with goblet-cell-like HT29-MTX cells [49]. However, the absence of other intestinal epithelial cells—such as Paneth cells and endocrine cells—along with differences in transporter expression and enzyme distribution can lead to inconsistent data compared to in vivo results [49].

Intestinal organoids have emerged as a promising next step in drug development; however, many models lack fluid flow, oxygen gradients, and mechanical forces necessary for physiological relevance. Gut-on-a-chip models can overcome these limitations by utilizing microfluidic platforms that enable gut-like functions. Cremer et al. reported an advanced in vitro model that mimics gut microbiota under physiological flow conditions [50]. By incorporating these additional layers of functionality, it may be possible to develop models that yield more reliable predictions regarding drug absorption and metabolism as well as disease pathology and developmental studies.

Liver-on-a-chip systems can replicate complex properties of the liver more effectively than hepatocyte organoids by addressing issues such as polarization deficiency while enhancing oxygen/nutrient supply and metabolic activity through shear force simulation. Using pumps and microfluidic channels allows for the recreation and maintenance of oxygen gradients and glucose concentrations while preserving appropriate cell-to-cell ratios and metabolite concentrations. Stern et al. recently reviewed the significance of OOC liver models for addressing idiosyncratic drug-induced liver injury (iDILI), which involves complex disease development processes that are generally challenging to replicate [51]. By incorporating non-parenchymal cells such as Kupffer cells, stellate cells, and dendritic cells into these model systems, hepatic inflammation—a key feature of iDILI—can be effectively modeled [51].

Haque et al. demonstrated that hepatocytes cultured within low-volume microfluidic channels maintained their differentiated state for up to 21 days compared to just 7 days in conventional cultures. Additionally, cells within these microfluidic environments produced hepato-inductive signals alongside pro-fibrogenic factors that sustained their phenotype and function [52].

A combined gut and liver-on-a-chip model can effectively mimic in vivo conditions, where there is a direct link between the absorption of drugs in the gut and their metabolism in the liver. Personalized models incorporating both gut and liver OOC systems can enhance toxicity studies, paralleling clinical trial conditions. Additionally, these models allow for the collection of cells or co-cultures for detailed genomic or proteomic analyses, providing further insights into toxicity profiles. The significance of growth factors and cytokines is often underestimated in many ex vivo disease or organ models. For example, growth factors such as platelet-derived growth factor (PDGF) and various cytokines have been shown to influence gastrointestinal inflammatory diseases and hepatic fibrosis. Therefore, including similar factors can help replicate physiological or pathophysiological conditions [53]. Regulating the supply of these factors will contribute to developing a sustainable OOC platform.

In addition to cell source, cell type, and circulating media, the extracellular matrix (ECM) used for cell support and the materials employed for the chip are critical determinants of successful OOC models. The priority lies in using a biocompatible ECM that sustains cells while maintaining their native physiological characteristics across multiple organs. Pairing this with a biocompatible chip interface, such as chitin or alginate, that possesses antibacterial and anti-inflammatory properties while facilitating the delivery of growth factors and cytokines can be highly beneficial [54].

Yang et al. reported on an integrated gut–liver-on-a-chip platform designed to model metabolic dysfunction-associated fatty liver disease (MAFLD) [55]. In this study, Caco-2 cells representing gut epithelial cells were connected via microfluidic channels to HepG2 cells mimicking liver cells. The gut–liver axis was shown to have a profound effect on MAFLD development. Free fatty acids (FFAs) were administered to the system over 1 and 7 days to initiate and progress MAFLD [55]. Furthermore, this model provided insights into MAFLD-like gene expression patterns, suggesting its potential use as an alternative to animal models in drug discovery (Figure 4).

### 3.5. Significance of Extracellular Matrix (ECM) on OOC and Similar Platforms

The ECM plays a major role in regulating cellular processes through structural support, biochemical signaling, and cellular signal regulation. It promotes healing and preserves tissue balance, which is important for the proper functioning of cells, tissue engineering, and developing ex vivo models [56]. Polysaccharide hydrogels are widely used to mimic the ECM due to their biocompatibility and similarity to the natural ECM. Tissue constructs can now be fabricated using controlled biomaterial composition, shape, and cellular components through 3D bioprinting. The hydrogels are sourced from diverse sources, including marine, plant/seed, starch, pectin, bacteria, and animal/human origins. They have differences in polysaccharide composition based on their types and origins and are being extensively characterized to improve tissue-like properties [56].

Similar to hydrogel-based ECM, a collagen-based scaffold has been used to model difficult-to-establish tumors for studying disease pathology and drug discovery. Miserocchi et al. reported a biomimetic collagen-based scaffold for oropharyngeal squamous cell carcinomas (OSCCs) and studied the impact of the ECM on the phenotypic and genotypic properties of the cells [57]. They also studied the impact of the scaffold on the cell line and primary cell responses to standard of care drugs [57]. De Vita et al. used collagen-based 3D models to characterize soft tissue sarcomas (STSs) and the activity of trabectedin on them [58]. They used primary cells derived from 10 patients and established a 3D culture system using collagen to analyze the activity of trabectedin on the cells. The models showed high morphological and genomic similarity to in vivo samples [58]. Broen et al. used primary human white adipose tissue sandwiched between adipose-derived stem cells to create a microphysiological system (MPS) to establish breast cancer models [59]. The sandwich model contained native ECM, mature adipocytes, fibroblasts, and immune cells and was stable for up to 14 days for studying cancer–ECM interactions and for drug-screening studies [59].

### 3.6. Recent Progress in Organ-on-a-Chip Platforms

Deng et al. recently published an extensive review on OOC platforms designed to model complex female reproductive pathophysiological conditions [60]. Compared to existing 2D models and in vivo systems, OOC platforms demonstrate the ability to replicate various physiological and pathophysiological states. Some notable models include placenta-on-a-chip, which represents placental barrier function; OOCs that study the invasive capabilities of placental trophoblast cells; and OOCs designed for investigating endometriosis and cancer [60].

Recent advances in 3D printing technology have significantly impacted the fabrication of microfluidic chips and tissue construction using bioinks. For instance, 3D printing technology has been explored for personalized treatments of Crohn’s disease [61]. Common materials used for 3D printing microfluidic chips include polydimethylsiloxane (PDMS), hydrogels, and polyethylene terephthalate (PET) [62]. The selection of materials is critical for establishing efficient OOC systems. Modified resins with no residual cytotoxicity, ideal transparency, and biocompatibility are employed in constructing OOC systems [63].

Many OOC projects utilize PDMS due to its cellular compatibility, high transparency, and low cost; however, it also presents drawbacks such as poor printability along with hydrophobic and porous characteristics. Villegas et al. demonstrated the application of a lubricant-infused coating on PDMS molds using fluorinated silane to reduce surface roughness and enhance optical properties [64]. Considerable progress has also been made with alternative materials like PET and thermoplastic polyurethane (TPU) for creating microfluidic chips through direct or indirect fabrication methods tailored to specific system needs.

Bioinks—comprising biomaterials, cells, and bioactive compounds—are essential for bioprinting tissues onto microfluidic chips in a process known as 3D bioprinting. Natural hydrogels commonly used as bioinks include gelatin, chitosan, and decellularized extracellular matrix-derived materials (dECMs) [62]. Widely adopted 3D bioprinting techniques encompass inkjet-based, extrusion-based, or digital light/laser-polymerization methods [65]. Cell sources can be derived from stem cells, such as induced pluripotent stem cells (iPSCs), cell lines, or cancer cells; diverse organ-specific OOC platforms have been established.

Kang et al. reported successful 3D bioprinting of liver OOC using extrusion bioprinting technology to construct hepatic lobular arrays that incorporate vascularization alongside various hepatic cell types [66]. The extruded hepatic lobules were functional while maintaining structural integrity, demonstrating potential applications in liver tissue engineering. Ma et al. showcased rapid 3D bioprinting of liver tissue, utilizing a hydrogel-based triculture that included hiPSC-derived hepatic cells along with vascular endothelial cells and adipose-derived stem cells [67]. This generated tissue exhibited high liver-specific gene expression levels alongside increased cytochrome P450 production and metabolite generation, underscoring its significance for drug screening and disease modeling experiments.

In addition to incorporating a vascular system into their models, Lee et al. integrated a biliary system into their bioprinted liver OOC platform. A decellularized ECM bioink was utilized to establish a 3D microenvironment featuring both vascular and biliary fluidic channels; this approach resulted in a model displaying liver-specific gene expression profiles and functional capabilities [68]. Furthermore, this model demonstrated effective drug responses, suggesting its potential utility for drug screening assays.

There has been enormous progress in the OOC platforms with several new organotypic models emerging [69]. Some of the major OOC models established for organ-level research include a second-generation lung-on-a-chip model [70]. The model reconstitutes the air–blood barrier using a biological, biodegradable, and stretchable membrane made of elastin and collagen, along with primary lung alveolar epithelial cells and primary lung alveolar endothelial cells. The generated biomembrane reproduced the composition and in vivo functionality of the lung alveolar barrier, and these properties were preserved for up to 3 weeks. Ross et al. described a novel microfluidic platform called the lymph node slice-on-a-chip to analyze local events in the lymph node, a structurally complex organ with high significance for human health. The lymph node slices maintained cellular organization, were accessible for local stimulation, and have the potential for the development of novel immunotherapies [71].

One of the premier studies in developing intestinal OOCs was conducted by Sung et al., who created the first 3D hydrogel-based platform to simulate human intestinal villi [72]. They utilized a combination of laser ablation and sacrificial molding techniques with calcium alginate to fabricate complex structures. Through this technology, they successfully produced microscale collagen structures that mimic human intestinal villi. In another significant study, Jalili-Firoozinezhad et al. established a complex human gut microbiome in an anaerobic intestine-on-a-chip [73]. The composition and function of the human microbiome, particularly the gut microbiome, have a profound impact on human health [74]. The microfluidic intestine OOC developed by Jalili-Firoozinezhad et al. allowed for the prolonged co-culture of human intestinal epithelium with stable communities of aerobic and anaerobic human gut microbiota. By creating a hypoxia gradient in the OOC, they were able to enhance barrier function and maintain a diverse microbial population, including over 200 unique taxonomic units from 11 different genera [73]. This OOC, which closely resembles the human gut, has the potential to aid in predicting drug absorption, developing probiotics, nutraceuticals, and microbiome-related therapeutics.

The filtration and reabsorption functions of kidney cells, or nephrons, consisting of the glomerulus, renal capsule, and renal tubule, were successfully modeled by several groups. Jang et al. established a kidney OOC lined by living human epithelial cells exposed to fluidic flow [75]. The microfluidic chamber mimics an in vivo system with a luminal channel and interstitial space, showing enhanced epithelial cell polarizations, primary cilia formation, increased albumin transport, and glucose reabsorption. Similarly, Sakolish et al. reported a kidney-based reusable microfluidic device [76]. The usage of human-induced pluripotent stem cells to successfully generate human islet organoids was reported by Tao et al. [77].

Zhang et al. reported a 3D tissue-engineered cell culture model on a chip for the heart, aimed at determining cardiac drug efficacy. This model utilized fibrin-based media and a high-speed impedance detection technique [78]. Schneider et al. demonstrated the use of heart OOCs with pluripotent stem cells [79]. The system is user-friendly and can generate multiple physiologically relevant tissues, with a simple readout using bright-field video microscopy. Some of the studies establishing OOC models are summarized below in Table 1.

Multi-OOC platforms involve cells from different organs cultured simultaneously and connected by a fluidic network. These platforms are useful for recreating the complexity, functions, and integrity of organ functions [80]. Tsamandouras reported a two-organ gut–liver OOC model using a novel mesofluidic multi-MPS platform coupled with computational modeling to connect a gut MPS with a liver MPS. A three-tissue OOC was reported by Skardal et al., where a liver, heart, and lung OOC was established with relative proportions [81]. The multi-organ OOC was used for drug and toxicology screening. Furthermore, multi-organ OOC models involving four to ten organs were reported [82,83].

## 4. Future Prospects

One of the major bottlenecks in the profitable commercialization of advanced in vitro models, such as spheroids or OOC platforms, is the lack of automated systems to support these processes. Recently, Lindner et al. demonstrated the development of an automated system utilizing affordable components, including a 3D bioprinter, to produce perfusion-ready OOC devices [84]. In comparison to existing alternatives like the BioAssembly Platform by Advanced Solutions Life Sciences and HUMIMIC AutoLab by TissUse GmbH, Berlin, Germany the system presented by Lindner et al. is notably adaptable, scalable, transferable, and customizable [85].

To establish robust hepatic screening platforms for assessing aflatoxin B1 toxicity, Schmidt et al. employed a mixture of gelatin, alginate, and Matrigel to maximize cell viability, printability, and stability over a duration of 21 days [86]. They utilized 3D-printed liver spheroids with in situ image-based quantification for screening drug-induced toxicity [81]. HepG2 liver spheroids were cultivated in mini-fabricated hydrogels using a 3D printer and analyzed for drug-induced mitochondrial permeability transition (MPT), apoptosis, and cytosolic calcium levels. Advances in imaging technologies combined with embedded sensors enhanced by AI/ML capabilities may facilitate the acquisition of multi-parametric data at higher resolutions for improved toxicity prediction.

Incorporating vascular networks into OOC platforms is critical for ensuring an adequate supply of nutrients and oxygen; however, it presents significant challenges. In liver OOCs specifically, adding bile-transporting microtubes or ducts has been shown to enhance physiological functions. Tronolone et al. employed deep learning techniques to analyze chemical and mechanical factors influencing MPS architecture in previously established models [87]. A chained neural network was trained using common morphological metrics to generate a vascular network quality index (VNQI), which can be applied across various MPS platforms. Furthermore, the VNQI algorithm was validated using vascularized islet chips.

Most OOC models rely on fluorescence image-based data outputs; however, data processing can be tedious, time-consuming, and prone to errors. Recent advances in AI technology now make efficient high-throughput data analysis feasible. A summary of recent reports on OOC or MPS platforms integrated with AI/ML technologies is provided below in Table 2.

Molecular mechanisms of drug toxicity include the generation of reactive oxygen species (ROS), apoptosis or necrosis pathway activation, alanine aminotransferase (ALT) secretion, and cytochrome P450 enzymatic activity, and metabolite profiling can be explored more for establishing predictive toxicity modeling. Iachetta et al. recently demonstrated a non-invasive optoporation-based chronic cardiotoxicity assessment using iPSC-derived cardiomyocytes [94]. Compared to traditional models, the optoporation-based method was able to read for up to 35 days in vitro [94]. Real-time monitoring of liver fibrosis development was carried out using embedded electrochemical sensors [95]. The microfluidic system was established using a 3D printer with TEER (trans-epithelial electrical resistance) and ROS sensors incorporated using chemical vapor deposition (CVD) and 3D printing technology. Liver fibrosis was induced by adding TGF-β1, resulting in the decrease in the TEER value and increase in ROS production. The liver MPS also helped to optimize ECM for ideal functioning. Similarly, Wang et al. reviewed kidney-on-a-chip models incorporating mechanical stimulation and integrated sensors, especially for measuring TEER [96]. Several molecular pathways contribute to the complexity of cancer, and tracking these complex pathways is cumbersome for establishing tumor-on-a-chip platforms. However, with the recent progress in the field of fluorescent nano- and microparticles capable of sensing diverse microenvironments, we can support developing efficient platforms. Fluorescent ratiometric nanoparticles are capable of sensing pH, oxygen content, ROS, and inorganic cations and anions like Ca^2+^, Na^+^, K^+^, and Cl^−^, and can serve as probes for biomarkers like matrix metalloprotease-2 (MMP-2) [97]. This fluorescent dye-based tracking can benefit not only the cancer field but also other therapeutic areas, including diagnostics platforms, fibrotic diseases, and inflammatory diseases. The merging of advanced molecular biology and biochemistry technologies, like multi-omics, including single-cell analysis, advanced imaging platforms, and mass spectrometry, is needed to track parameters like metabolites, cellular phenotype, and cell health in real time to develop efficient and predictable in vitro models.

## 5. Conclusions

Drug discovery is a lengthy and costly process, traditionally relying on animal models for toxicity and efficacy studies, which significantly contribute to drug failures due to their inherent limitations. However, recent advancements in in vitro models, particularly organoids and OOC systems, coupled with 3D bioprinting and AI/ML technologies, represent a significant leap forward in this field. These innovative platforms offer the potential to closely mimic human physiological and pathophysiological conditions, thereby enhancing the relevance of toxicity and efficacy studies while minimizing reliance on animal testing. Given that the majority of drugs are administered orally, developing robust gut–liver OOC models that accurately reflect physiological aspects, mechanistic inputs, and human-relevant cellular phenotypes can have a substantial impact on drug discovery processes by providing insights that align more closely with human biology. Despite challenges associated with incorporating complex features such as vascular networks, extracellular matrix components, and standardization of OOC systems, ongoing research continues to address these issues [98]. The successful application of deep learning techniques to analyze microphysiological system architecture, along with advances in sensing molecular pathways involved, exemplifies the progress being made in this area. As we move toward a future where personalized medicine becomes increasingly attainable, the development of advanced in vitro models will play a crucial role in transforming drug discovery into a more efficient, ethical, and effective process. Regulatory measures, patient consent, data security, and safety measures should be incorporated before implementing AI technologies on patient samples to create personalized OOC models [99]. Strict and far-reaching policies with proper training datasets to avoid bias and discrimination will minimize the issues associated with AI and can increase patient confidence to pursue similar research in an efficient way [100].

## Figures and Tables

**Figure 1 cimb-47-00007-f001:**
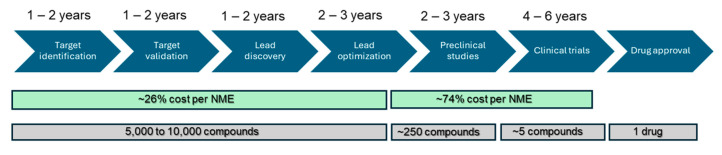
Schematic representation of the drug discovery and development pipeline. The first step is target identification, followed by target validation. Lead discovery is the next step and is generally pursued using high-throughput screening. Lead optimization is carried out before preclinical studies. The drug candidates selected from preclinical studies are taken to clinical trials, and a successful drug is approved for clinical usage.

**Figure 2 cimb-47-00007-f002:**
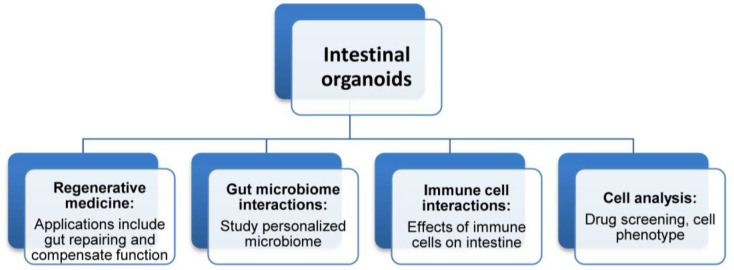
Uses of intestinal organoids. Intestinal organoids have a variety of common applications, including replacing or compensating for functional structures, mimicking host–microbe interactions, studying interactions with immune cell types, and investigating drug toxicity or developmental biology.

**Figure 3 cimb-47-00007-f003:**
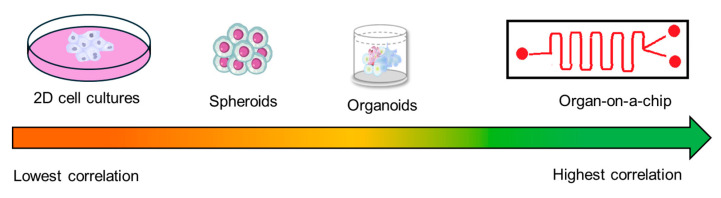
Clinically relevant in vitro models that demonstrate increased correlation to human biology. This progression is evident through the advancement from 2D cell culture models to 3D models like spheroids or organoids and ultimately to organ-on-a-chip models.

**Figure 4 cimb-47-00007-f004:**
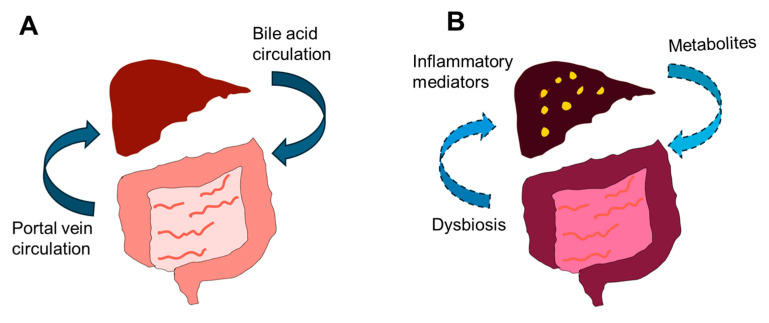
Gut–liver axis during normal conditions compared to conditions with metabolic dysfunction-associated fatty liver disease (MAFLD). (**A**) In normal conditions, there is bidirectional communication between the intestines and liver, known as the gut–liver axis, which helps maintain homeostasis. (**B**) However, under MAFLD conditions, factors originating in the gut, such as dysbiosis, leaky gut, and inflammatory cytokines, can impact the liver. Additionally, altered bile acid metabolites can affect gut function. An efficient organ-on-a-chip model can be used to represent this scenario [50].

**Table 1 cimb-47-00007-t001:** Some of the organ-on-a-chip studies summarized below with the ECM/platforms used.

Sl No.	Organ Focused	ECM/Platform Used	Reference
1	Lung	Elastin and collagen membrane	[70]
2	Lymph node	Tissue slice-on-a-chip	[71]
3	Intestine	Hydrogel-based	[72]
4	Gut (anaerobic microbiome)	Matrigel and collagen	[74]
5	Kidney	Microfluidic flow systems	[75,76,77]
6	Heart	Fibrin/impedance/microscopy	[78,79]

**Table 2 cimb-47-00007-t002:** Examples of organ-on-a-chip integrated with artificial intelligence for data analysis in various applications.

Sl No.	Platform Tested	Target/Therapeutic Area	Reference
1	Microscopic images	Red blood cell deformability	[88]
2	2D time-lapse imaging	cancer cell cycle dynamics, motility, etc.	[89]
3	Lens-free imaging and quantitative agglutination	SARS-CoV-2 sensing	[90]
4	Label-free efficacy testing using tumor spheroids	Tumor spheroids for cancer	[91]
5	Live cell imaging	Brain metastasis	[92]
6	Smartphone-based imaging	Fentanyl detection	[93]

## Data Availability

No new data were created or analyzed in this study. Data sharing is not applicable to this article.

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
