# Peer review of "Advanced In Vitro Models for Preclinical Drug Safety: Recent Progress and Prospects"

_cimb, 2024, doi:10.3390/cimb47010007_

Round 1
Reviewer 1 Report
Comments and Suggestions for Authors
Advanced in vitro Models for Preclinical Drug Safety, Recent Progress and Prospects
Suggestion: major revisions
1. Overall, your text lacks strong evidence for your statements and needs more references for your claims. This is consistent right from the introduction until the conclusion. Please ensure to add more references and ideally, as a thumb rule, a good review article has about 100 references.
2. Please include a timeline/figure indicating the complexity of drug discovery in terms of how long it takes, what are its different stages, how expensive it gets with every stage etc.
3. For section 2 – please add relevant studies with animal models. While human-relevant data is the ultimate goal, animal models have provided significantly towards our current understanding towards pharmacology. Rather than saying that we are better off without them, please acknowledge their contribution and provide details of the most relevant studies in the last 5 years.
4. Please provide significantly more references in section 3. Currently, all your statements are simply claims with no supporting evidence. Especially for SAR and ADME – there is plenty literature out there! Please work through this, throughout the document.
5. Please provide a table for section 3.5 with studies in OOC in the last 5-10 years that are most relevant to you. There are plenty studies that have indicated advancement that had nothing to do with AI. This will indicate the growth of OOCs before AI was a possibility.
6. Please add a paragraph outlining the limitations that one may face with the additional use of AI in OOCs. Reflect on how you will protect patient identity and confidentiality when using primary cells for OOCs, how will you optimise and manage the use of the enormous amount of data generated etc.
Author Response
Dear Reviewer 1,
Thank you for your thoughtful and constructive feedback on our manuscript entitled "Advanced in vitro Models for Preclinical Drug Safety, Recent Progress and Prospects." We appreciate the time you took to review our work and provide valuable insights. Below, we address each of your comments in detail.
- Overall, your text lacks strong evidence for your statements and needs more references for your claims. This is consistent right from the introduction until the conclusion. Please ensure to add more references and ideally, as a thumb rule, a good review article has about 100 references.
We acknowledge your concern regarding the lack of strong evidence for our statements throughout the paper. In response, we have conducted a thorough literature review and incorporated additional references to substantiate our claims. We understand the importance of a well-supported narrative and have aimed to increase our reference count significantly towards the target you suggested.
- Please include a timeline/figure indicating the complexity of drug discovery in terms of how long it takes, what are its different stages, how expensive it gets with every stage etc.
We agree that a visual representation of the drug discovery process would enhance clarity. Therefore, we have included a timeline figure outlining the various stages of drug discovery, including duration and cost implications associated with each phase. This addition provides readers with a clearer understanding of the complexities involved.
- For section 2 – please add relevant studies with animal models. While human-relevant data is the ultimate goal, animal models have provided significantly towards our current understanding towards pharmacology. Rather than saying that we are better off without them, please acknowledge their contribution and provide details of the most relevant studies in the last 5 years.
We appreciate your suggestion to include relevant studies involving animal models. While we maintain that human-relevant data is crucial for future advancements, we recognize the historical significance of animal models in pharmacology research. Accordingly, we have added a section acknowledging their contributions and included several pertinent studies from the last five years.
- Please provide significantly more references in section 3. Currently, all your statements are simply claims with no supporting evidence. Especially for SAR and ADME – there is plenty literature out there! Please work through this, throughout the document.
Your feedback regarding the need for more references in Section 3 has been taken seriously. We have revisited this section and enriched it with additional literature supporting our statements on Structure-Activity Relationships (SAR) and Absorption, Distribution, Metabolism, and Excretion (ADME). This enhancement aligns with best practices for scientific discourse.
- Please provide a table for section 3.5 with studies in OOC in the last 5-10 years that are most relevant to you. There are plenty studies that have indicated advancement that had nothing to do with AI. This will indicate the growth of OOCs before AI was a possibility.
To address your request effectively, we have included key studies related to Organ-on-Chip (OOC) technology over the past 5-10 years that are relevant to our discussion and created a comprehensive table summarizing—specifically highlighting advancements that occurred independently of AI developments.
- Please add a paragraph outlining the limitations that one may face with the additional use of AI in OOCs. Reflect on how you will protect patient identity and confidentiality when using primary cells for OOCs, how will you optimise and manage the use of the enormous amount of data generated etc.
Lastly, we appreciate your insight into discussing limitations associated with AI integration within OOCs. A discussion has been added in Conclusion section to reflect on potential challenges such as patient identity protection and data management strategies when utilizing primary cells in OOCs.
We believe these revisions strengthen our manuscript significantly and align it more closely with scholarly standards while addressing your concerns comprehensively.
Thank you once again for your valuable input; it has greatly contributed to enhancing our work. We look forward to any further comments or suggestions you may have.
Sincerely,
Dileep Nair and Ralf Weiskirchen

Reviewer 2 Report
Comments and Suggestions for Authors
The authors Nair and Weiskirchen provide an overview of the preclinical models for drug safety.
This is a useful review, well organized.
The manuscript would benefit from the followings:
1. a graphical abstract depicting all the models should included in the manuscript
2. A section including 3D scaffold model should be added to the review. In this regard a variety of 3D scaffold has been proposed. The authors should include the following reference for proper discussion:
Three-dimensional collagen-based scaffold model to study the microenvironment and drug-resistance mechanisms of oropharyngeal squamous cell carcinomas. Cancer Biol Med. 2021 Mar 27;18(2):502–16. doi: 10.20892/j.issn.2095-3941.2020.0482. Epub ahead of print. PMID: 33772505; PMCID: PMC8185858.
Recent advances in 3D bioprinted polysaccharide hydrogels for biomedical applications: A comprehensive review. Carbohydr Polym. 2025 Jan 15;348(Pt B):122845. doi: 10.1016/j.carbpol.2024.122845. Epub 2024 Oct 16. PMID: 39567171.
The potential role of the extracellular matrix in the activity of trabectedin in UPS and L-sarcoma: evidences from a patient-derived primary culture case series in tridimensional and zebrafish models. J Exp Clin Cancer Res. 2021 May 11;40(1):165. doi: 10.1186/s13046-021-01963-1. PMID: 33975637; PMCID: PMC8111914.
High-throughput screening of more than 30,000 compounds for anthelmintics against gastrointestinal nematode parasites. bioRxiv [Preprint]. 2024 Oct 31:2024.05.16.594481. doi: 10.1101/2024.05.16.594481. PMID: 39554023; PMCID: PMC11565780.
3. Some example of co-culultre system should be included. i.e.
Tumor-Stroma Crosstalk in Bone Tissue: The Osteoclastogenic Potential of a Breast Cancer Cell Line in a Co-Culture System and the Role of EGFR Inhibition. Int J Mol Sci. 2017 Jul 29;18(8):1655. doi: 10.3390/ijms18081655. PMID: 28758931; PMCID: PMC5578045.
Minor revisions are requested
Author Response
The authors Nair and Weiskirchen provide an overview of the preclinical models for drug safety.
This is a useful review, well organized.
Dear Reviewer 2,
Thank you for your positive feedback on our manuscript entitled "Advanced in vitro Models for Preclinical Drug Safety, Recent Progress and Prospects." We are grateful for your constructive suggestions that will enhance the quality of our review. Below, we address each of your comments:
The manuscript would benefit from the followings:
- a graphical abstract depicting all the models should included in the manuscript
We appreciate your suggestion to include a graphical abstract depicting the major models discussed in our manuscript. We have created an informative graphical abstract that visually summarizes the drug development along with various in vitro models and their human correlation, which we believe will aid readers in grasping the key concepts more effectively.
- A section including 3D scaffold model should be added to the review. In this regard a variety of 3D scaffold has been proposed. The authors should include the following reference for proper discussion:
Three-dimensional collagen-based scaffold model to study the microenvironment and drug-resistance mechanisms of oropharyngeal squamous cell carcinomas. Cancer Biol Med. 2021 Mar 27;18(2):502–16. doi: 10.20892/j.issn.2095-3941.2020.0482. Epub ahead of print. PMID: 33772505; PMCID: PMC8185858.
Recent advances in 3D bioprinted polysaccharide hydrogels for biomedical applications: A comprehensive review. Carbohydr Polym. 2025 Jan 15;348(Pt B):122845. doi: 10.1016/j.carbpol.2024.122845. Epub 2024 Oct 16. PMID: 39567171.
The potential role of the extracellular matrix in the activity of trabectedin in UPS and L-sarcoma: evidences from a patient-derived primary culture case series in tridimensional and zebrafish models. J Exp Clin Cancer Res. 2021 May 11;40(1):165. doi: 10.1186/s13046-021-01963-1. PMID: 33975637; PMCID: PMC8111914.
High-throughput screening of more than 30,000 compounds for anthelmintics against gastrointestinal nematode parasites. bioRxiv [Preprint]. 2024 Oct 31:2024.05.16.594481. doi: 10.1101/2024.05.16.594481. PMID: 39554023; PMCID: PMC11565780.
Your recommendation to include a section on 3D scaffold models is well taken. We have added a dedicated section discussing various types of 3D scaffolds and their relevance in drug safety studies. Furthermore, we have incorporated the references you provided into this discussion to ensure comprehensive coverage of recent advancements in this area.
- Some example of co-culture system should be included. i.e.
Tumor-Stroma Crosstalk in Bone Tissue: The Osteoclastogenic Potential of a Breast Cancer Cell Line in a Co-Culture System and the Role of EGFR Inhibition. Int J Mol Sci. 2017 Jul 29;18(8):1655. doi: 10.3390/ijms18081655. PMID: 28758931; PMCID: PMC5578045.
Thank you for emphasizing the importance of co-culture systems. We have included examples relevant to tumor-stroma interactions and their implications in pharmacology, specifically citing the study you mentioned regarding breast cancer cell lines and their osteoclastogenic potential. This addition enriches our review by illustrating how co-culture systems contribute to understanding complex biological interactions.
We believe that these revisions significantly enhance our manuscript and address your valuable suggestions effectively. Thank you once again for your insightful feedback; it has played an important role in improving our work.
We look forward to any further comments or suggestions you may have.
Sincerely,
Dileep Nair and Ralf Weiskirchen
